# Institutional decarbonization scenarios evaluated against the Paris Agreement 1.5 °C goal

Robert J. Brecha [1,2,3 ✉], Gaurav Ganti [1,4 ✉], Robin D. Lamboll [5], Zebedee Nicholls [6,7,8,9], Bill Hare[1], Jared Lewis[7,8,9], Malte Meinshausen [6,7,8], Michiel Schaeffer [1,10], Christopher J. Smith [9,11] & Matthew J. Gidden[1,9]

Scientifically rigorous guidance to policy makers on mitigation options for meeting the Paris Agreement long-term temperature goal requires an evaluation of long-term global-warming implications of greenhouse gas emissions pathways. Here we employ a uniform and transparent methodology to evaluate Paris Agreement compatibility of influential institutional emission scenarios from the grey literature, including those from Shell, BP, and the International Energy Agency. We compare a selection of these scenarios analysed with this methodology to the Integrated Assessment Model scenarios assessed by the Intergovernmental Panel on Climate Change. We harmonize emissions to a consistent base-year and account for all greenhouse gases and aerosol precursor emissions, ensuring a self-consistent comparison of climate variables. An evaluation of peak and end-of-century temperatures is made, with both being relevant to the Paris Agreement goal. Of the scenarios assessed, we find that only the IEA Net Zero 2050 scenario is aligned with the criteria for Paris Agreement consistency employed here. We investigate root causes for misalignment with these criteria based on the underlying energy system transformation.

[1] Climate Analytics, Berlin, Germany. [2] Hanley Sustainability Institute, University of Dayton, Dayton, OH, USA. [3] Renewable and Clean Energy Program, University of Dayton, Dayton, OH, USA. [4] Geography Department, Humboldt-Universität zu Berlin, Berlin, Germany. [5] Center for Environmental Policy, Imperial College London, London, UK. [6] Australian-German Climate and Energy College, The University of Melbourne, Melbourne, Australia. [7] School of Geography, Earth and Atmospheric Sciences, The University of Melbourne, Melbourne, Australia. [8] Climate Resource, Melbourne, Australia. [9] International Institute for Applied Systems Analysis, Laxenburg, Austria. [10] The Global Center on Adaptation, Rotterdam, The Netherlands. [11] Priestley International Centre for Climate, University of Leeds, Leeds, UK. ✉email: robert.brecha@climateanalytics.org; gaurav.ganti@climateanalytics.org

Since its adoption in 2015, governments, international agencies and private entities have increasingly recognized the implications of the Paris Agreement's 1.5 °C long-term temperature goal (LTTG) for greenhouse gas emissions reduction planning in both the near- and long-term. Governments have submitted or are preparing updates of their Nationally Determined Contributions (NDCs) and are encouraged to submit long-term low greenhouse gas development plans (Article 4 of the Agreement[1]), aimed at aligning short- and long-term strategies. The foundations on which country targets are based are guided, directly or indirectly, by a variety of sources of information judged to be authoritative, including scientific research institutes[2], international agencies, or private companies. Importantly, such authoritative sources also affect planning and decision making by investors[3] who aim to anticipate climate policies, and their decisions in turn can drive or hold back setting ambitious emissions reduction targets. We can observe this influence, for example, in the ruling of the Hague District Court, which directed Royal Dutch Shell (Shell) to reduce (net) emissions by 45% below 2019 levels, by 2030[4]. The ruling draws on scenarios from the International Energy Agency (IEA) and Shell, in addition to mitigation pathways underlying the Intergovernmental Panel on Climate Change (IPCC) Special Report on 1.5 °C (SR1.5)[5].

Skea et al. propose a categorization of energy system scenarios in terms of being outlooks, exploratory, or normative[6]. Outlooks are short- to medium-term projections or forecasts, whereas exploratory scenarios represent a range of plausible futures. Normative scenarios are explicitly associated with the achievement of a desired end state (in this case, a temperature goal) and have primarily been the subject of investigation by the Integrated Assessment Modeling (IAM) community. Most IAMs represent different greenhouse gases and aerosol precursors over a long-term horizon (2100), a necessary characteristic for assessing the consistency with the Paris Agreement LTTG. A large number of emission mitigation pathways generated by IAMs have been previously assessed and categorized with respect to their climate outcomes in SR1.5[5,7]. The SR1.5 database also included a small selection of non-IAM pathways (including two scenarios from the IEA, and one from Shell). Whereas the climate outcomes of the IAM pathways were assessed by the SR1.5 author team, the temperature statements of the non-IAM pathways were self-assessed.

Assessing if a given emissions mitigation pathway (here, we use the term scenario synonymously) adheres to the Paris Agreement requires some historical context of the climate negotiation process. The Cancun Agreement had previously established a goal of limiting global temperature to "below 2 °C"[8], which was interpreted by the scientific community as a 66% probability (or "likely" chance) of maintaining this limit[9]. Article 2.1.a of the Paris Agreement strengthens this target to "holding the increase in the global average temperature to well below 2 °C above pre-industrial levels and pursuing efforts to limit the temperature increase to 1.5 °C", while Article 4.1 requires parties to "achieve a balance between anthropogenic emissions by sources and removals by sinks of greenhouse gases"[1]. The strengthened temperature goal therefore requires a substantially higher margin and likelihood of holding warming below 2 °C, for example the "very likely" level of 90% probability of not exceeding 2 °C (approximately equivalent to at least a 33% probability of not exceeding 1.5 °C), as well as achieving net-zero emissions in this century[10,11]. Further, pathways highlighted in the Summary for Policymakers (SPM) of SR1.5 are "as likely as not" to limit warming to 1.5 °C by the end of the century (i.e., with 50% probability)[12]. Based on the scenarios underlying the IPCC's Sixth Assessment Report (AR6), three criteria for Paris Agreement compatibility of pathways are established[13]: (1) never have a

greater than 66% probability to overshoot 1.5 °C and achieving 1.5 °C at the end of century with at least a 50% chance, (2) always hold warming below 2 °C with a 90% chance or higher, and (3) achieve net zero greenhouse gases in the second half of the century. Here we adopt this approach with a slightly more relaxed criteria (2) of a close to 90% chance to categorize pathways as Paris Agreement compatible—this is aligned with the median outcome of the C1 category of pathways in the IPCC report (see footnote 48 in the AR6 WG III SPM: "Scenarios in this category are found to have simultaneous likelihood to limit peak global warming to 2C throughout the 21st century of close to and more than 90%"[14]).

In this work we present a transparent temperature assessment of institutional decarbonization scenarios (i.e., normative scenarios modelled by organizations that have historically been most associated with either outlook or exploratory scenarios, and we identify scenarios from institutions including Shell[15], BP[16], the IEA[17,18], and Equinor[19]) on an equal footing with IAM scenarios. We also demonstrate explicitly how the challenges to such an assessment can be resolved, and implement a framework to analyze the climate outcomes of these scenarios. We further assess key underlying energy system features that drive emissions pathways, thus providing an evaluation of the structural dynamics that lead a given scenario to satisfy (or not) the Paris Agreement LTTG.

## Results

**A framework to assess the climate impact of institutional scenarios.** We identify three challenges to understanding the stated climate outcome of published institutional emission pathways (see the scenario claims outlined in Table 1). These include the time horizon of the scenarios, the limited representation of greenhouse gases and aerosol emissions, and perceived inconsistency and relative opacity of the climate assessments. The temporal scope of the Paris Agreement necessitates both near-term (i.e., peak warming) and long-term (i.e., end-of-century warming) evaluation to assess compatibility with the LTTG. Since most published institutional scenarios, whether outlook, exploratory or normative (the Sky 1.5 scenario from Shell is an exception) do not extend beyond mid-century, the first challenge is to use a consistent methodology to extend pathways to the year 2100.

The second key challenge is that most institutional scenarios focus on $CO_2$ emissions from the energy sector (and sometimes include industrial process emissions). To evaluate the temperature outcome, a representation of all greenhouse gases and aerosol precursor emissions is needed, including non-energy $CO_2$, emissions from land use, land use change and forestry, and non-$CO_2$ emissions (from methane, for instance). Some institutional scenarios, such as the IEA NZE scenario include some discussion of these emissions but do not report detailed data in their publicly available scenario data, preventing a thorough comparison. The Sky 1.5 scenario from Shell is the only scenario we assess here that presents detailed trajectories for all major greenhouse gases or presents any data on aerosol precursor emissions (although the only aerosol precursor it includes is SO2).

The final challenge identified here is transparency in the quantification of the climate impact of the scenario. There are three manifestations of this problem in the scenarios we assess. First, some scenarios (including the IEA's 2020 Sustainable Development Scenario (SDS), and the BP scenarios) make references to temperature outcomes that are difficult to trace back to a concrete assessment. This is outlined in the second column of Table 1, which, for these scenarios is the only

**Table 1 Institutional scenario characteristics—claims, time horizon and gas coverage.**

| Institution (Scenario) | Scenario claim | Scenario endpoint | Gas coverage |
|---|---|---|---|
| Equinor[19] (Rebalance) | The stated cumulative emissions starting from 2018, through to 2050, are assumed to be 740 Gt $CO_2$ in this scenario and this is claimed to be consistent with holding warming well below 2 °C. | 2050 | Energy $CO_2$ |
| Shell[15] (Sky 1.5) | The carbon budget for this scenario is assessed to be relatively higher than IPCC estimates (747 Gt $CO_2$), yet it is still claimed to be consistent with holding warming to 1.5 °C in 2100 with a high temporary overshoot. | 2100 | Energy $CO_2$ Industrial Process $CO_2$ AFOLU CO2 $CH_4$ N2O HFCs PFCs $SF_6$ |
| BP[16] (Rapid) | This scenario is assessed to be consistent with scenarios that hold warming well below 2 °C in 2100. The assessment is performed by comparing the relative reduction in $CO_2$ emissions from energy use. | 2050 | Energy $CO_2$ |
| BP[16] (Net Zero) | This scenario is assessed to be consistent with scenarios that hold warming to 1.5 °C. The assessment is performed by comparing the relative reduction in $CO_2$ emissions from energy use. | 2050 | Energy $CO_2$ |
| IEA[17] (SDS) | The IEA presents a conditional assessment for the warming outcome of the scenario. If emissions stay at zero after 2070 the scenario is claimed to hold warming to 1.65 °C or less and this is assessed to be consistent with holding warming well below 2 °C. If net-negative emissions are deployed, warming under the SDS is assessed to be below 1.5 °C with a 50% chance in 2100. | 2040 | Energy and Industrial Process $CO_2$ |
| IEA[18] (NZE) | The IEA presents a conditional assessment for the warming outcome of the scenario. Assuming a proportional reduction in non-$CO_2$ emissions, the scenarios is assessed to be consistent with a 50% chance of holding warming below 1.5 °C without a temporary overshoot. | 2050 | Energy and Industrial Process $CO_2$ |

information that we are aware of, presented to support the statements regarding scenario consistency with the Paris Agreement.

Second, some scenarios present carbon budget constraints (i.e., the maximum amount of anthropogenic $CO_2$ that can be emitted, while still achieving a desired temperature goal, over a desired timeframe) that vary widely for the same temperature goal; for example, the Equinor Rebalance scenario and the Shell Sky scenarios report carbon budgets larger than 700 Gt$CO_2$, but only Shell claims to achieve the 1.5 °C goal (see Table 1 and page 93 at ref. [15]).

Third, even where a climate model is used to assess the climate outcomes (e.g., Shell Sky), the comparability with the simplified climate model parameters used in the IPCC assessments is limited. For an assessment performed in line with the SR1.5 assessment (see "Methods"), a complete set of emission species is necessary. We document the necessary gases in Table 1 of the Supplementary Information.

Given the relative opacity in the self-assessed climate impact of institutional scenarios, we propose a consistent set of steps to perform such an assessment, in a manner that allows for direct comparison with IAM scenarios (Fig. 1). We briefly outline the steps here, with further details presented in "Methods". We first harmonize all the emissions to the same historical dataset, and then proceed to check if the scenarios extend until 2100—if not, we extend the data from the last available year until 2100 using the Constant Quantile Extension (CQE) method[20]. The CQE method extends an emission trajectory by determining the position (quantile) of the last reported data point with respect to a distribution of emissions scenarios that extends to the targeted end year (here, those from the SR1.5 set of scenarios), and then using the same position for future time steps. The CQE method has previously been applied to extend the emission levels implied by the Nationally Determined Contributions (NDCs) that are defined until 2030[21,22]. Adding to the existing publications of

the method, we study its robustness in further detail in Supplementary Information Section 1.

We then proceed to infer missing emission species using the Quantile Rolling Windows (QRW) infilling method, a previously documented quantile regression technique[23] (summarized in the "Methods" section), that finds the median expected value of the infilled emission given the known energy and industrial $CO_2$ emissions. Where a scenario provides a discussion of missing emission species in the report, but does not report a time series in the public data, we still select this method as a default for reasons of transparency.

The impact of alternative infilling methods can be found in Supplementary Table 2. The resulting multi-gas emission trajectories are provided as an input to the reduced complexity coupled carbon cycle and climate model MAGICC6[24] in its probabilistic setup, in line with the approach adopted by SR1.5 (see "Methods"). We use MAGICC6 here, to allow for direct comparison with the SR1.5 climate outcome assessment. The scenarios are then classified using the same categorization scheme as in SR1.5, as provided in Table 3 in the Supplementary Information. Any reduced complexity climate model (e.g., FaIR[25,26]) which implements the openscm interface can be similarly applied, and we include results obtained by using FaIR with its SR1.5 configuration in Supplementary Information Section 5.

**Evaluating the multi-gas emission trajectories**. The key variable for each scenario is $CO_2$ emissions from energy and industrial processes. Comparing the values for this key variable in the SR1.5 database and the institutional scenarios provides a first-order approximation of the climate implications of the institutional scenarios (Fig. 2a). The two pathway classes in the SR1.5 that meet the warming limit of the Paris Agreement result in 2030 emission levels of 13.6 Gt$CO_2$ [13.2–16.1 interquartile range] for

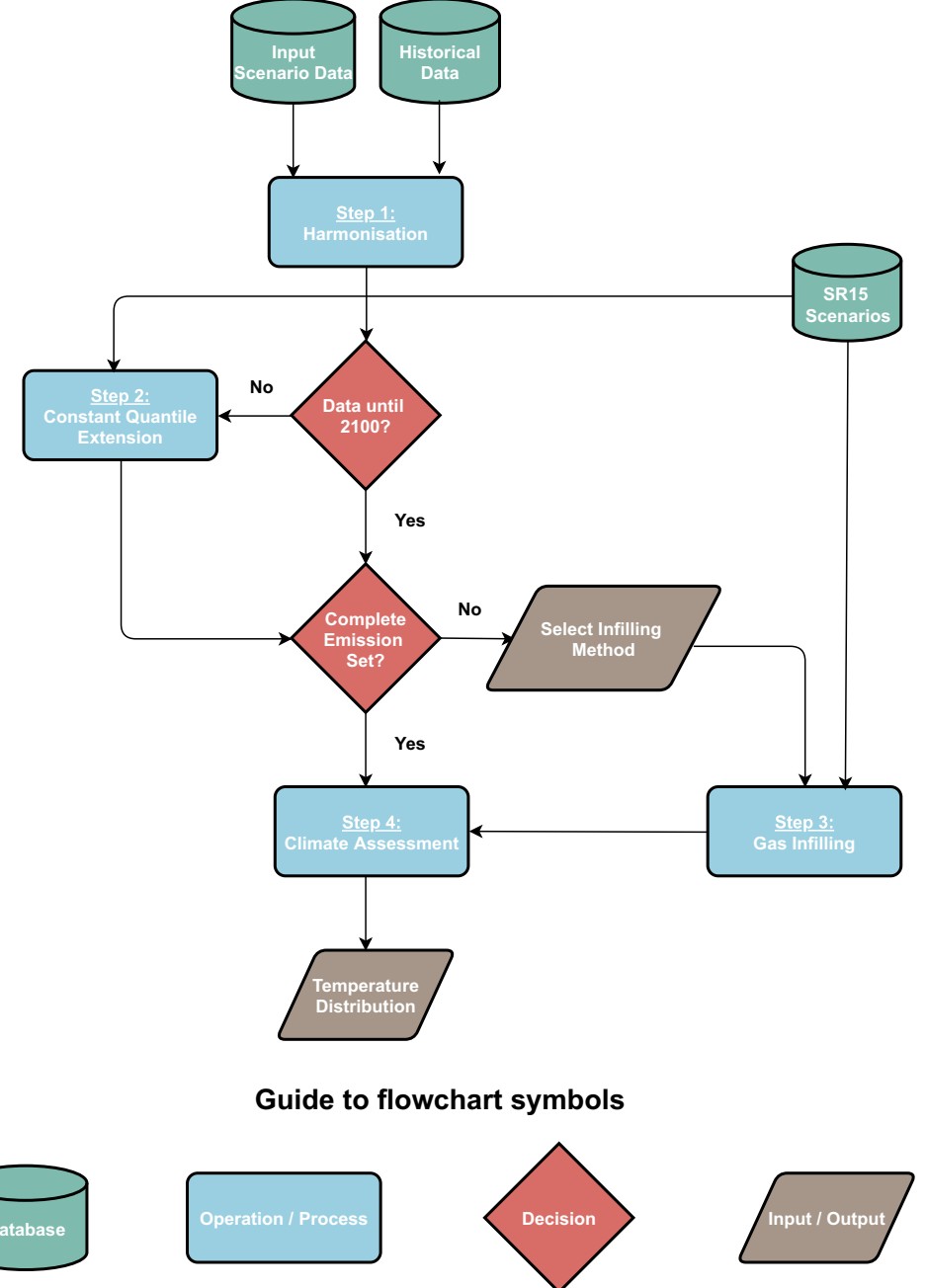

**Fig. 1 Schematic diagram of the assessment framework.** The input scenario data refers to the institutional scenario data assessed in this study. The schematic is composed of database inputs (institutional scenario data, data from the scenarios underlying the IPCC's Special Report on 1.5 °C), data processing steps, decision steps that lead to the application of certain steps, as well as user-defined inputs, and outputs of the framework.

the Below 1.5 °C (henceforth referred to as no-overshoot pathways) category and 21 GtCO$_2$ [18.6–22.6 interquartile range] for the 1.5 °C low-overshoot category. Apart from the IEA NZE scenario (that is close to the median of the low-overshoot pathways), all other institutional scenarios assessed here are either above the interquartile range of the low-overshoot pathways or, in the case of Shell's Sky 1.5 scenario, outside the range of the low-overshoot pathways. By 2050, the IEA NZE scenario lies below the median of the low-overshoot pathways, and the Shell Sky 1.5 scenario remains above the low-overshoot pathways as well as the interquartile range of the 1.5 °C high-overshoot (henceforth referred to as high-overshoot pathways) pathways.

The reported energy and industry CO$_2$ emissions have implications for the infilled greenhouse gases, notably for CO$_2$ emissions from Agriculture, Forestry and Land Use (AFOLU), CH$_4$, and N2O emissions. Methane emissions (Fig. 2c) reach 2030 levels of 156 Mt CH$_4$ (129–248 interquartile range) for the no-overshoot pathways and 236 Mt CH4 (189–257 interquartile range) for the low-overshoot pathways. In comparison, the Sky 1.5 scenario from Shell reaches a 2030 level of 426 Mt CH$_4$, which is above the interquartile range of even the high-overshoot scenarios. A similar characteristic is seen for N$_2$O emission as seen in Fig. 2d. We explore the reasons for sensitivity to infilling methods and related analytical considerations in further detail in Supplementary Information Section 2.

CO$_2$ emissions from AFOLU (Fig. 2b) show a large variation in the SR1.5 pathways. Most infilled scenarios cluster in the interquartile range of the high-overshoot and 2 °C scenarios,

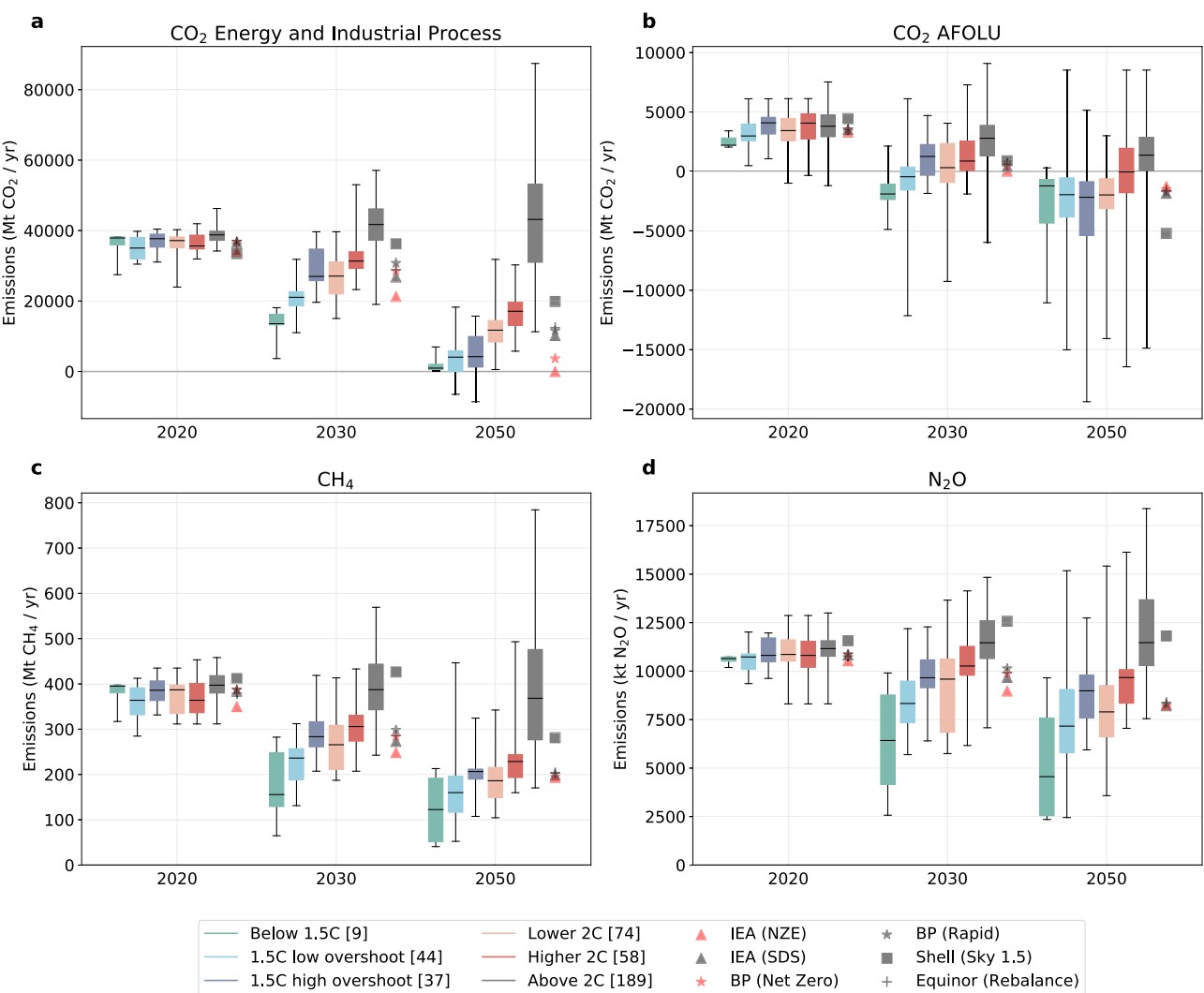

**Fig. 2 Comparison of emission characteristics between the SR1.5 pathways and the institutional scenarios assessed in this study. a** $CO_2$ emissions from energy and industrial processes, (**b**) $CO_2$ emissions from Agriculture, Forestry, and Land Use (AFOLU), (**c**) $CH_4$ emissions, (**d**) $N_2O$ emissions. The box represents the interquartile range with the median represented by the solid horizontal line. The whiskers represent the full range across the corresponding pathway class. All emissions (apart from panel a) are infilled using the Quantile Rolling Windows (QRW) method, except for Shell Sky 1.5 which reports these emissions.

with the reported 2050 value from Shell a high outlier and implying a heavy reliance on land-based CDR that goes well beyond most of the low-, no- and high-overshoot pathways (in 2050, it is at the 22nd percentile of the joint distribution of these three pathway categories). Due to the overlapping ranges between the different pathway classes (e.g., high-overshoot characteristics overlap with the low-overshoot characteristics), it is not sufficient for a pathway to be located in the low-overshoot $CO_2$ emission range to assess its compatibility with the Paris LTTG.

**Climate categorization and properties of scenarios.** In Fig. 3a, d, and Supplementary Table 4, we show that only four scenarios (IEA Net Zero, BP Net Zero, IEA SDS, and Shell Sky) meet the first climate outcome necessary for Paris Agreement consistency (i.e., balance between sources and sinks in the second half of the century). For non-$CO_2$ GHGs we aggregate the infilled gases for each scenario, converting non-$CO_2$ emissions to $CO_2$ equivalent emissions using Global Warming Potential 100 (GWP100) from the IPCC's 4th Assessment Report[27]. The choice of the greenhouse gas accounting metric affects the relative weighting of

different short- and long-lived gases and the timing of net zero greenhouse gas emissions[28]. Here, we select GWP100, as it is the default reporting metric under the Paris Agreement, and an interpretation of the goals of the Paris Agreement using GWP100 is internally consistent[29].

Figure 3b, e shows the 21st century temperature trajectories for the six scenarios considered here, to assess which scenarios meet the other two climate outcomes indicated earlier in this paper. Most of the scenarios we assess here overshoot the 1.5 °C warming limit by a significant margin (shown in Fig. 3c as an exceedance probability, with the corresponding climate outcome thresholds indicated as grey horizontal lines). Equinor's Rebalance scenario peaks at a median warming of 1.73 °C above pre-industrial (here 1850–1900 is used as a proxy for pre-industrial) levels in 2060, and a similar margin of overshoot is observed in BP Rapid (1.73 °C in 2058), Shell Sky (1.81 °C in 2069) and the IEA SDS (1.78 °C in 2056). All of these scenarios would be classified as Lower 2 °C pathways and are inconsistent with the Paris LTTG, failing to hold warming well below 2 °C (nor pursuing 1.5 °C). BP's Net Zero scenario results in a median end of century warming of 1.5 °C (consistent with the scenario claim),

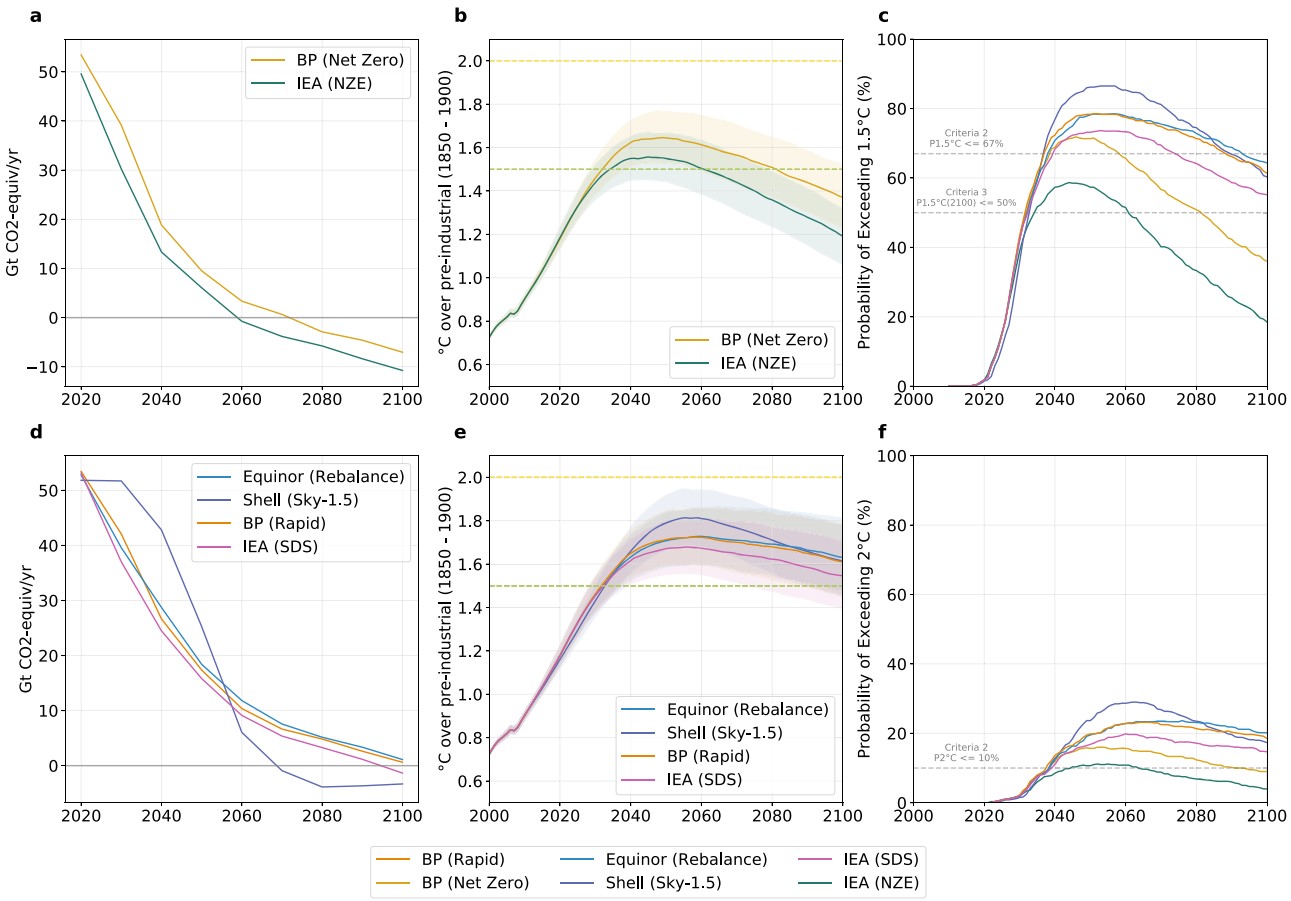

**Fig. 3 Assessment of the three criteria for Paris Agreement consistency. a, d** Greenhouse gas emissions. **b, e** Temperature rise above 1850–1900 (the solid line is the ensemble median and the shaded plumes are the 33rd–66th percentile). **c** Probability of exceeding 1.5 °C. **f** Probability of exceeding 2 °C. Calculations for (**b, c, e, f**) are performed using MAGICC6.

**Table 2 Key climate outcomes for assessed institutional pathways using MAGICC with the Quantile Rolling Windows (QRW) infilling method.**

| Source | Scenario | Median Level of Peak Warming | Median Year of Peak Warming | SR1.5 Climate category | P1.5 °C Max (2100) | P2 °C Max (2100) |
|---|---|---|---|---|---|---|
| Equinor | Rebalance | 1.73 °C | 2060 | Lower 2 °C | 78% (64%) | 23% (20%) |
| Shell | Sky | 1.81 °C | 2059 | Lower 2 °C | 86% (60%) | 29% (17%) |
| BP | Rapid | 1.73 °C | 2058 | Lower 2 °C | 78% (61%) | 23% (18%) |
| BP | Net zero | 1.65 °C | 2049 | 1.5 °C high overshoot | 71% (36%) | 16% (9%) |
| IEA | SDS | 1.68 °C | 2056 | Lower 2 °C | 73% (55%) | 19% (14%) |
| IEA | NZE | 1.56 °C | 2045 | 1.5 °C low overshoot | 58% (18%) | 11% (4%) |

The SR1.5 categories are based on the criteria discussed in the text. The final two columns provide the maximum and end-of-century (in parentheses) exceedance probabilities of 1.5 °C and 2 °C.

the high temporal overshoot (median peak warming of 1.65 °C) results in the pathway being classified as a 1.5 °C high overshoot pathway. The one exception to note is the recently-released IEA NZE scenario which we assess as a 1.5 °C low-overshoot scenario. This scenario has the lowest maximum peak exceedance probability, and we assess it to be close to meeting the condition of being very likely less than 2 °C (>90% likelihood). It also achieves a balance between sources and sinks in the second half of

the century. Table 2 summarizes the key temperature indicators for the six scenarios.

A key potential source of uncertainty, even with our internally consistent method, is in regards to the climate categorization due to different infilling methods. Among the five scenarios with a large number of gases infilled (Shell is omitted here), four are situated in the same climate category irrespective of the infilling method (see Supplementary Table 5). One scenario (IEA NZE)

would be situated in a higher climate category (1.5 °C high overshoot) when the RMS method is applied—we trace this back to the specific characteristics of the pathway selected to infill under the RMS method. This uncertainty can be reduced by more complete multi-gas modeling, with non-$CO_2$ emissions assessed in a manner consistent with the actual production of fossil fuels, biofuel production, among others. We also explore the difference in results between MAGICC6 and FaIR (the second simple climate model used in the SR1.5 assessment) in Supplementary Information Section 5, and our results agree with the SR1.5 assessed differences in climatic outcomes between the models. With updated model versions, and calibrations that will be used in the upcoming WGIII contribution to the IPCC's 6th Assessment Report, the simple climate models are likely to show a higher degree of agreement on the magnitude of warming[30].

**Comparing high-level mitigation levers**. Whether a given scenario will achieve the aim of the Paris Agreement is a strong function of the underlying energy system transformation. Warszawski et al.[31] propose a set of mitigation levers that can be used to compare the mitigation options selected by different pathways[31]. These include CDR deployment, changes in carbon intensity of final energy ($CIt$), change in final energy demand ($Et$) and relative reduction in non-$CO_2$ emissions (all reductions relative to a base year, which the authors select as 2018). We connect to that work by selecting 2010 as the base year for comparison in line with our historical harmonization year (see "Methods"), and examine the indicators $CIt$ and $Et$ in 2030 and 2050 with respect to that base year to evaluate some energy system mitigation characteristics (Supplementary Table 6). We do not include scenarios from BP in this comparison because they do not report Final Energy in the data they make available. We only compare two indicators since we have used the infilling methods to complete the set of greenhouse gases in the institutional pathways, and it would then be inconsistent to evaluate quantities resulting from our own infilling method as non-$CO_2$ and non-energy-system emissions levers.

Final energy and the carbon intensity of that energy behave independently as levers that can both be used to reduce emissions. For example, the Equinor Rebalance and the IEA NZE scenarios have final energy demand consistent with the range of 1.5 °C compatible SR1.5 scenarios at about 90% of the 2010 level, illustrating the increasing importance of energy efficiency, whereas Shell Sky 1.5 has a demand 50% higher than the SR1.5 scenarios. On the other hand, the carbon intensity in Rebalance is much higher, as is that of Shell Sky 1.5, in line with higher 2 °C pathways and approximately 40% of the 2010 $CI$, rather than 5–10% as in SR1.5 PA-compatible scenarios. Only the IEA NZE scenario has both $Et$ and $CIt$ consistent with low-overshoot pathways both in 2030 and in 2050.

**Comparing technology preferences across the pathways**. The speed and depth of transformation in the electricity sector provide a significant indication of the potential achievement of a temperature goal, given that all other energy sectors depend to a greater or lesser degree on electrification. In IAM scenarios there is a clear relationship between the share of coal that remains in electricity generation and the temperature categorization, although in all cases coal generation decreases strongly (Fig. 4c). The Equinor and Shell scenarios tend to match more closely the remaining coal generation of the higher-temperature IAM scenarios; the BP Net Zero scenario has more coal generation in 2030 than the PA-compatible IAMs, although it converges to those scenarios by 2050. This latter feature indicates a somewhat slower transition away from coal and helps explain the

categorization as non-PA-compatible. The IEA NZE scenario shows the most similar behavior to the IAM scenarios that are PA compatible, most closely matching low-overshoot scenarios in 2030 and then below-1.5 °C scenarios in 2050.

In the case of natural gas, results shown in Fig. 4d indicate that those scenarios closest to Paris Agreement compatibility also tend to be those with the most rapid decrease in natural gas shares in electricity generation, including the IEA NZE scenario. However, the range of use of natural gas is large, depending on scenario and model, and therefore reflects the uncertainty in the literature as to the bridging role for natural gas in the power sector[32–34]. Part of this uncertainty is due to the potential in some models for significant fossil-fuel carbon capture and storage potential. Given that levelized costs of solar PV and onshore wind are already cheaper in many regions of the world than new fossil-fuel-based electricity generation even without the additional costs of CCS, economic decisions about low-carbon sources are likely to favor the former[35].

IAM scenarios with low- or no-overshoot tend to have lower levels of wind and solar share in electricity generation in 2050 than do the IEA NZE, Shell Sky and BP Net Zero scenarios, as shown in Fig. 4a (~45% vs. 60%–65%), possibly reflecting a historical tendency of IAMs to underestimate the potential for higher shares of variable renewable energy[36–40]. All models tend to show similar characteristics for nuclear power by mid-century (Fig. 4b), with a share in electricity generation somewhat lower than at present, but higher in absolute terms due to the increasing contribution of electricity in the energy system (Fig. 4e). The role of nuclear power varies more between IAMs than it does as a function of scenario, as seen in the large spread in Fig. 4b, but with relatively consistent median values. Final energy either decreases or grows only slightly by mid-century in 1.5 °C compatible scenarios (Fig. 4f).

## Discussion
The Paris Agreement sets not only a long-term temperature goal, but also intermediate conditions constraining allowable temporary overshoot of 1.5 °C. More recent literature also introduces a further bound on the energy system transformation through sustainability limits on the potential for deployment of bioenergy with carbon capture and storage (BECCS) and of carbon-dioxide removal (CDR)[41–48]. Pathways that delay reductions in fossil-fuel consumption in the near-term and thus lead to a high overshoot of 1.5 °C run the risk of counting on an over-dependence on CDR later in the century. Given that the analysis presented in this paper infers $CO_2$ emissions (and, implicitly, removals) from AFOLU, and implicitly infers (using the gas infilling methods highlighted earlier) the non-AFOLU $CO_2$ removals for most of the scenarios, we have not presented an assessment of the CDR deployment in institutional pathways – however, further transparency in this regard (as we observe in the IEA Net Zero scenario data), where emission reductions and removals are reported separately would important. In addition, anthropogenic greenhouse gas emissions beyond $CO_2$ must be reduced as well; in some cases, such as with some methane emissions, this will tend to occur along with the reduction in combustion or recovery of fossil fuels. However, emissions from agriculture, such as methane and N2O will require mitigation as well.

Since normative scenarios relevant to the Paris Agreement and published by institutions such as the IEA and fossil-fuel companies provide important input to policymakers and investors, they should provide a complete pathway to the end of the century for all GHG emissions for all sectors so that temperature assessments can be made. A methodology for evaluating total GHG and aerosol precursor emission pathways is presented here

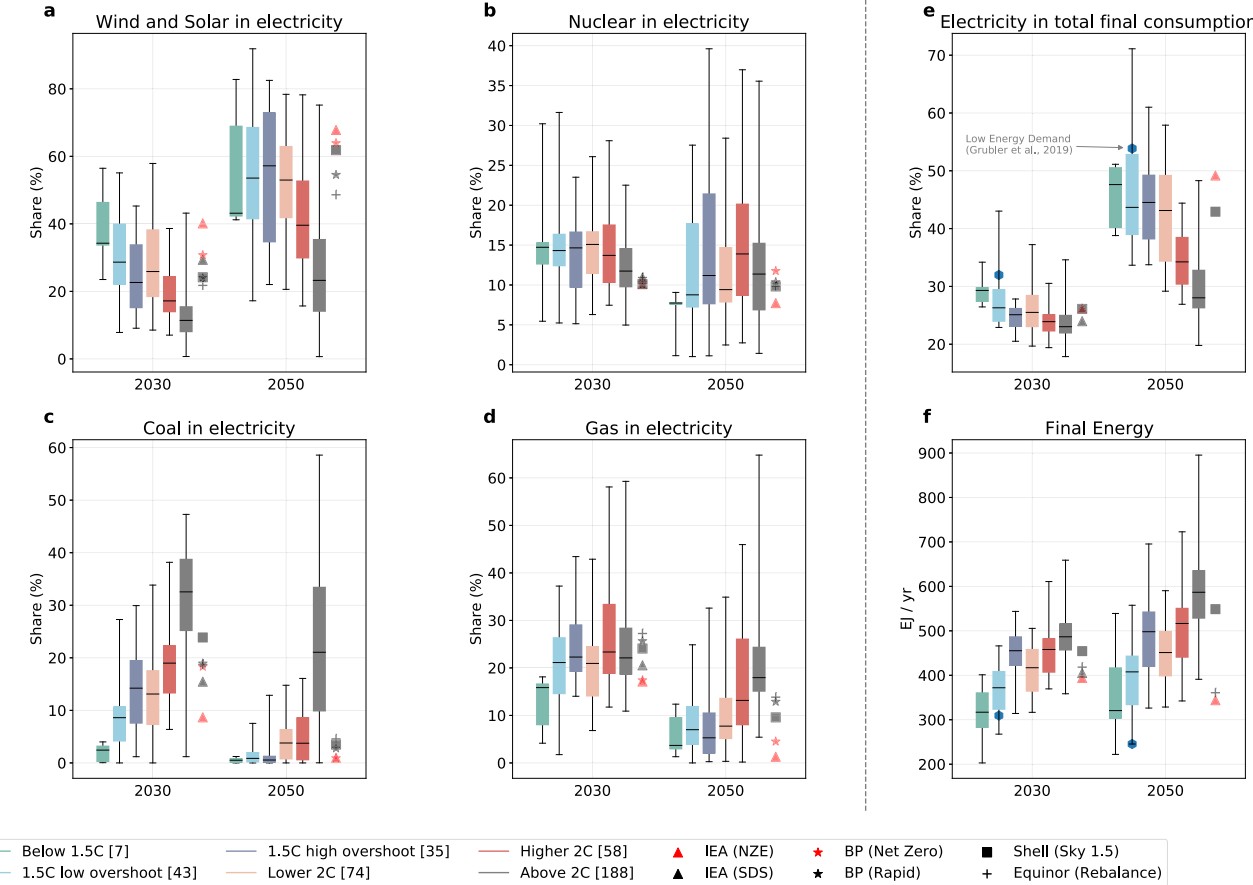

**Fig. 4 Key energy system characteristics across pathways.** The dashed vertical line separates two groups of indicators. Panels (**a–d**) are focused on the share of different fuels and/or technologies in total electricity generation. Panels (**e, f**) capture total final consumption, and the electrification of end-use. **a** Share of wind and solar in electricity generation. **b** Share of nuclear in electricity generation. **c** Share of coal in electricity generation. **d** Share of natural gas in electricity generation. **e** Share of electricity in total final consumption. **f** Total final energy consumption. The numbers in parentheses refer to the number of IPCC's Special Report on 1.5 °C (SR1.5) database scenarios of each category. Where a specific scenario is missing in the panel (for instance, in panel **e**), it is either because the data are not reported, or, in the case of the International Energy Agency's (IEA) Sustainable Development Scenario (SDS) from 2020, because the scenario ends in 2040.

in a manner that allows intercomparisons even in the absence of such provided pathways, while acknowledging the uncertainties inherent in using such techniques. Published institutional pathways that do not actually lead to the LTTG of the Paris Agreement will likely provide a misleading view of the transformations needed for reducing GHG emissions both in the near-term and the long-term.

For the most part, the institutional pathways analyzed here do not achieve the Paris Agreement LTTG, or do so with substantial interim overshoot. Primarily, this is due to a continued reliance on fossil fuels that is greater than IAM pathways that achieve the PA LTTG. For example, although the use of coal shows a steep decline in all pathways, it is notable, and of particular importance for policymakers and for investment decision making, that the role of natural gas is less clear, demonstrating a large range of uncertainty in the various pathway categories. In general, though, pathways that achieve the PA LTTG without significant overshoot do not appear to allow for a bridging role for natural gas; however, this is an area ripe for further investigation[49–51]. On the other hand, some of the institutional pathways indicate the potential for higher and faster penetration of renewable energy uptake than do many IAMs, also an important signal for discussions about meeting the PA LTTG.

Our focus has been on institutional pathways, but similar limitations are valid for the growing literature of bottom-up

energy modeling approaches that find the potential for 100% renewable energy-based systems by mid-century[52–61], which tend to outpace estimates of renewable penetration rates compared to IAMs[36,62,63]. A claim of 100% renewable energy by 2050 may align with energy sector benchmarks for PA-compatibility, but it is not sufficient to guarantee these pathways meet the LTTG. They should also be self-consistently evaluated by including full GHG pathways. The trend in the scientific community is towards full data and model transparency, an increasingly important part of the science-policy interface. In the case of claims on pathway compatibility with international climate agreements, this transparency should extend to the data and assumptions required to confirm such statements.

## Methods

**Data Sources and Handling.** In this study, we construct a harmonized dataset of institutional scenarios and compare them with the scenarios underlying the IPCC Special Report on 1.5 °C (SR1.5) as described in the following section 64,65,66,7. We assess normative scenarios (i.e., scenarios that claim to reach a Paris Agreement future) from four institutions: Equinor, the International Energy Agency, Shell and BP. For scenarios where data for 2020 are not available, we extend the data series to 2020 based on the historical trend observed in the data reported by the study. We note that this does not account for the effect of COVID-19 on emissions; however, the IEA projects a strong growth in emissions in 2021 and hence the bias induced by this assumption (in one year) is unlikely to affect our assessment. To fill in missing gases, we use $CO_2$ emissions from energy and industrial processes as the lead gas. Scenarios from Equinor and BP do not report industrial process

emissions. For these scenarios we assume that industrial process emissions follow the same trajectory as the IEA Sustainable Development Scenario (SDS) from 2020[17]. We note that a newer version of the IEA SDS has been published in 2021 that extends until 2050[64].

**Harmonization**. Emissions are harmonized in line with the approach adopted in SR1.5, using historical data from the SSP database[65–67] and harmonized scenario data from the SR1.5 scenario database hosted by IIASA[7]. We harmonize all emission trajectories to the 2010 values from our historical dataset using the harmonization package aneris[68]. We select a ratio harmonization method, with the ratio (between the historical data and the scenario data in the base year) converging to 1 in the last available scenario year. For the Shell scenarios that report data until 2100, we use the default method of aneris (reduce ratio to 2080), which is the default adopted in the CMIP6 emission harmonization routines[69].

**Extending Series Until 2100**. Published scenarios from IEA, Equinor, and BP have data to 2040 or 2050. We extend these data from the last available scenario data point to 2100 using the Constant Quantile Extension (CQE) method[20] and implemented in the Python package Silicone[70]. We first identify the position (quantile) of the scenario emissions in the last available year with respect to an underlying distribution of emission pathways. We then apply the same quantile to the emission distribution to extend the scenario emissions until 2100. In this paper the underlying emission distribution is drawn from the database of scenarios underlying the Special Report on 1.5 °C[7,71]. With the CQE, we attempt to capture some element of the underlying model dynamics while extending the pathway. A drawback of this method is that the underlying distribution of emissions may not represent structural transition in the energy system models (used by the three institutions) appropriately. To evaluate the validity of the method, we assess its effectiveness in reconstructing known data (here, data from the SR1.5). We truncate each pathway (i.e., model and scenario combination) in the SR1.5 database at 2050 and then use the CQE method to extend the pathways to 2100. We then calculate the root mean square difference between the original value and the extended value in each time step, normalized by the spread of values in that time step, defined by

$$\varepsilon = \Sigma_i \sqrt{\Sigma_t \frac{\left(p_{i,t} - q_{i,t}\right)^2}{n_t \sigma_t^2}/n_i} \qquad (1)$$

where $\varepsilon$ is the error, $p_{i,t}$ is the CQE-extended value of pathway $i$ at time $t$, $q_{i,t}$ is the originally projected value at that time, $n_{i(t)}$ is the number of pathways (times) being summed over and $\sigma_t$ is the standard deviation of original projections at that time. This effectively compares the error when using the CQE method with the error from using the average value in the database at that time—a value of one corresponds to the error in using the average value in the database at each time. The results of this method being applied to various emissions types can be found in Supplementary Table 1. For $CO_2$ emissions from energy and industrial processes, the error measured this way is 0.22 indicating that this method, for the dataset considered here, is far better than simply using the average value from the database.

**Gas Infilling**. Gas infilling (i.e., inferring missing emission species) is necessary to construct a complete, multi-gas emissions trajectory that can be used to assess the climate impact of a scenario. In this paper, we apply the infilling approaches implemented in the Python package Silicone[70]. The key premise of the infilling techniques is that there is a relationship between the emissions that are represented in the scenario and emissions that are to be inferred—the different infilling methods correspond to different ways of defining that relationship. In the main body of this paper, we use the Quantile Rolling Windows (QRW) technique to infer missing emission species. We use $CO_2$ emissions from energy and industrial processes (Emissions|$CO_2$|Energy and Industrial Processes) as the lead gas consistently. We provide a short description of the QRW infilling method here, as a summary of the published methodology[70]. Like all quantile regression methods, QRW considers a scatter plot of two variables and considers how to draw a continuous line through them such that a given fraction of points will be below the line—in our case, 50%, as we want to find the median. In the QRW method, evenly spaced points across the lead variable (x-axis) are chosen and at each point a weight of

$$\frac{1}{\left(1 + (\text{lead value difference})^2\right)} \qquad (2)$$

is applied to all data points, with the normalization of the lead value difference affecting how sensitive to local changes the relationship is. Then, the weighted median of the follower values (y-axis) at these points is calculated. Finally, we interpolate between these points to determine our regression line. A higher number of evaluation points enables more complicated relationships, with more changes in gradient between the lead and follower to be resolved, but also increases the risk of overfitting to the data—we use the default 11 points. The QRW technique gives the best balance between robustness to small changes and accuracy in infilling results. We use it here in preference to the RMS closest technique, which is slightly more accurate when applied to the SR1.5 database but more variable, and as a whole-

pathway technique means that the values at one time are influenced by values at another. This latter feature is best avoided when we are using techniques to project emissions forward in time from an earlier stage. For those scenarios that report 2020 emissions that account for the impact of the COVID-19 pandemic, we include the reported 2015 value to reduce the bias introduced by interpolating between 2010 and 2020 (this applies to Shell Sky and IEA NZE). We assess sensitivities to two other common infilling methods that have been used in the literature (the 'Equal Quantile Walk' and 'RMS closest' methods).

**Climate assessment**. To assess the climate impact of the scenarios, we provide the constructed multi-gas emission pathways as an input to the reduced complexity carbon cycle and climate model MAGICC6[24] using a Python-based wrapper Pymagicc[72] to process the data. The probabilistic distribution of climate impacts is assessed using 600 sets of parameters that reflect the climate sensitivity range assessed by the IPCC in the 5th Assessment Report and the Special Report on 1.5 °C, as well as to represent carbon cycle uncertainties. Updated distributions, in line with the 6th Assessment Report of the IPCC or more recent literature may lead to different conclusions about the climate implications of these scenarios. We calculate the temperature rise relative to the 1986–2005 mean value and add 0.61 °C to make the comparison relative to the 1850–1900 reference level. This follows the approach from AR5[73] (Chapter 6, Figure 6.12–6.13) that was subsequently used in SR1.5. The categorization follows the categories in SR1.5 that are described in further detail in Supplementary Table 3.

**Reporting summary**. Further information on research design is available in the Nature Research Reporting Summary linked to this article.

## Data availability
The input institutional scenario data that supports the findings of this study are available on request from the corresponding authors due to license restriction from the data providers.

## Code availability
The software and scripts that we used to perform this analysis are publicly available at: https://gitlab.com/gaurav-ganti/institutional_scenarios.

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

## Acknowledgements

R.J.B. acknowledges support from the EU Horizon 2020 Marie-Curie Fellowship Program (H2020-MSCA-IF-2018, proposal number 838667—INTERACTION). G.G., M.G., M.S., B.H. acknowledge support German Federal Ministry for the Environment, Nature Conservation and Nuclear Safety (grant no. 16_II_148_Global_A_IMPACT). RDL acknowledges support from funding from the EU Horizon 2020 research and innovation program under grant agreement No 820829 (CONSTRAIN). C.J.S. was supported by a NERC/IIASA Collaborative Research Fellowship (NE/T009381/1). We would like to thank Daniel Huppmann for developing the SR1.5 python notebooks, Yann Robiou du Pont for reviewing an earlier draft of this manuscript, and Shivika Mittal for discussions on transparency and data issues in IEA scenarios.

## Author contributions

These authors contributed equally: R.J.B., G.G. R.J.B., M.J.G., B.H., and M.S. conceived the research and analysis. G.G., R.J.B., and Z.N. carried out the analyses. G.G., R.D.L., Z.N., and C.J.S. performed sensitivity and robustness analyses of results. M.M. leads the development of MAGICC and created the climate parameter configuration used in this study. Z.N., R.D.L., J.L., and M.J.G. developed and maintain software utilized by the methodology used in this study. R.J.B., G.G., and M.J.G. led the drafting of the manuscript, with input and feedback from the author team.

## Competing interests

The authors declare no competing interests.
