## [Peer Review File · Nature Communications]

Institutional Decarbonisation Scenarios Evaluated Against the Paris Agreement 1.5°C GoalREVIEWER COMMENTS

Reviewer #1 (Remarks to the Author):

This paper compares and discusses institutional scenarios as compared to the IPCC SR1.5 scenario database. The paper is well written and structured and the methodology is well explained. Although the article is relevant its contribution to the literature in the field is very limited, especially for a nature paper.

Specific comments:

in paragraph of line 74: there are 3 categories mentioned and only two are explained.

The authors mention the carbon the carbon budget concept (paragraph of lines 137) but do not define carbon budget. Although the reader assumes the same calculation as the SR1.5 was taken, this should be clearly stated. e.g. From which to which year and which gases and sectors are included.

Please discuss better the reason behind the statement in lines 106.

The authors seem to consider IPCC and MAGICC6 as the gold standard of climate temperature calculations. While this is, generally, accepted in this literature, in order to make strong statements about institutional scenarios the authors should dedicate some discussion on why this model is better than any other that may have been used.

Still related to this point, in "Any reduced complexity climate model (e.g., FaIR21) which implements the openscm interface can be similarly applied." Why this sensitivity was not performed? This would help the robustness of the statements made.

In Table S2, the row of "Energy and Industrial Process CO₂ (Mt CO₂ / yr)" is redundant as no infill method was used.

In Figure S1: should be given also in co₂ equivalent in order to better compare the impact in terms of warming contributions. The caption should mention what are the gray lines.

The authors decide on "no overshoot" term, which is miss leading and not coherent with the legends in the figures nor the results of figure 3.

I suggest a line on the zero in Figure 2. Also include the reduction relative to start year or the carbon budgets.

A figure and discussion on total GHG would greatly help.

Although institutional scenarios probably not report on aerosol, a sensitivity on this is also required. Even if very simple, for example using the maximum reduction achieved with aerosol removal from the SR1.5 or SSP database (SSP may be better because it has different aerosol removal pathways).

Still related to figure 2, it would also add transparency to add which share of the emissions where calculated with the infill method.

In "heavy reliance on land-based CDR that goes well beyond most of the low- and high overshoot pathways." Instead of "most" better to mention the quantile.

In fig 4 total electricity is clearly not the best indicator to show.

I guess the number in brackets in the figures is the number of scenarios but this needs to be written in the caption. Please revise all the captions of your figures and make sure all the information is described in there.

Still related to Figure 4: without any indicator on negative emissions this figure may be very misleading. Same way the conclusion may also be less robust.

Figure 4 seems to indicate that the institutional scenarios assume a very high energy demand that is not foreseen by the models. Some models may reduce energy demand in the light of a temperature target, whereas institutionally scenarios foresee a much higher demand regardless of a climate constraint. This may be an important point of distinction between these scenarios. A discussion on total energy demand assumed would help the comparison (maybe taking only SSP5 like scenarios one would find the same ranges as the institutional ones).

Reviewer #2 (Remarks to the Author):

This study develops and applies a framework for examining whether alleged Paris-aligned climate scenarios of institutions such as Shell and IEA really are Paris-aligned. The study is timely, well-written and has the potential for a large impact, also beyond academia. I have a bunch of comments that mostly relate to clarity of presentation. I will be happy to recommend the study for publication after the authors have considered my (mostly minor) comments.

My two only semi-major comments are that it is not easy to grasp the QRW infilling method (see detailed comments to line 129 and 508 below) and that the three criteria for being Paris-aligned that are presented in the introduction do not appear to be fully applied to evaluate the institutional scenarios later in the study (see detailed comment to line 184).

Detailed comments:

- Line 18: If you have room in the abstract, I think you could broaden the appeal of your study by naming a couple of the institutions involved, such as Shell and International Energy Agency.
- Line 39: As an example of the impact of institutional scenarios, you may want to refer to the recent ruling by The Hague District Court over Shell, where the court made several references to its Sky 1.5 scenario and its alleged alignment with the Paris goal:
<https://uitspraken.rechtspraak.nl/inziendocument?id=ECLI:NL:RBDHA:2021:5339>
- Line 74: I encourage you to define what you mean by “outlook” and “exploratory” scenarios, as you do it for “normative” scenarios. I see that you partially do that around line 85, but I think the text might work better if you define all three terms upfront. Also, it is currently not clear from the text if you use “outlook” and “exploratory” interchangeably or if they mean different things.
- Line 87-89: I am not sure I understand this sentence. What does large uncertainties relate to? Who are proposing the alternative scenarios (the creators of the outlook scenarios or you?). Consider rewording the sentence.
- Line 99-101: What do you mean by “detailed greenhouse gas emission pathway”? Does it relate to a comprehensive coverage of GHGs and sources (as suggested by Table 1)? Please clarify.
- Line 117: It looks like the first step in Figure 1 is harmonization of historical and scenario data. Consider creating a better alignment between the figure and the text, for example by noting the step numbers on the framework figure.
- Line 119-121: I was not familiar with the CQE method beforehand and had to consult your Methods section to understand the essence of it. I think this sentence could be improved if you add a bit of information, e.g.,: “with respect to a distribution of reference emissions scenarios that extends to the desired endpoint”. You could also consider mentioning that you used the scenario set from the Special Report on 1.5°C when applying the CQE method.
- Figure 1: Consider stating in the caption what the information in parentheses (Silicone, Aneris, etc.) refer to or delete it if it is not essential for the framework. Also, consider adding, to the captions, what the four different shapes/colours refer to.
- Line 129: Consider briefly (1-2 sentences) explaining the QRW infilling method, as you did for the CQE method.
- Line 132-134: From reading this, I get the impression that you used the same modelling approach to assess the temperature outcomes of scenarios as in SR1.5? If so, consider stating that more explicitly. Also, please clarify what you mean by “classified”?

- Table 1: The reference to the QRW infilling method seems out of place in the table caption. Why not mention “using MAGICC6”, as in Figure 3, instead?
- Figure 3: To create better alignment with the text and Table 1, you can consider showing the four pathways that you categorized to Lower 2°C in Figure 3a and the other two (better performing) pathways in Figure 3b.
- Line 184: In the beginning of your study (lines 41-55), you present three climatic outcomes that scenarios should meet to be considered “Paris Agreement compatible”. However, in this paragraph, it appears that you are only addressing one of those three (whether scenarios are “very likely” to not exceed a peak warming of 2 degrees). For completion and consistency, I encourage you to explicitly comment on how the institutional scenarios perform relative to the other two criteria (“as likely as not” to limit warming to 1.5°C by the end of the century and achieving a balance between sources and removals). You could also indicate two of the criteria on Figure 3c (below 50% by 2100) and 3d (below 10% at all times).
- Line 200: perhaps something like “high-level mitigation levers” would be better than “technology-agnostic mitigation levers” in the heading. Otherwise, the choice of the “technology-agnostic” term only really makes sense after the reader has made it to the next section...
- Line 210-212: If I understand correctly, you are here arguing why you did not consider CDR deployment and relative reduction in non-CO2 emissions? If so, I think you could present the argument more explicitly.
- Figure 4: The sequence of the charts does not match the figure caption (or the body text).
- Line 447-449: I am not sure I understand. Does this, for example, mean that the emissions in all years of a scenario are downscaled a bit if the scenario’s 2010 emissions are a bit higher than actual 2010 emissions? Consider rephrasing. ♦ After reading further, I see that the harmonization approach is detailed in lines 461-466. It may create a better reading experience if you only describe the harmonization approach once.
- Line 449: Missing left-hand parenthesis?
- Line 454: Please provide a reference to the mentioned IEA projection.
- Line 464-465: Should it be “the default method of anemis” (its not like the software is an actor that can select things, right?)?
- Line 508: It would be helpful if you could give one or more short examples of how a certain characteristic of the trajectory of CO2 emissions from energy and industrial processes (the lead gas) results in a certain characteristic of the infilled gas trajectory, according to the QRW method. For example: Does a relatively high value for the lead gas also mean a relatively high value of the inferred gas? Does a steep reduction of the lead gas result in a proportionally steep reduction of the inferred gas? Or is the relationship more complicated?

Please see our itemized responses to the comments by both reviewers. We would like to thank the reviewers for the thoroughness of their review of our original manuscript. The original comments are in black, our responses are in red

REVIEWER COMMENTS

Reviewer #1 (Remarks to the Author):

This paper compares and discusses institutional scenarios as compared to the IPCC SR1.5 scenario database. The paper is well written and structured and the methodology is well explained. Although the article is relevant its contribution to the literature in the field is very limited, especially for a nature paper.

Thank you for your comments on the relevance, and structure of the paper. We identify the following contributions that this paper makes to the literature:

- For the first time, an assessment of the climatic outcomes of key scenarios used by policy makers, institutional bodies (e.g., the IEA and OECD), and civil society is provided. These scenarios play a critical part in the policy creation process, benchmarking what policies and efforts are needed to limit global warming to specific, legally defined, levels. To date, these scenarios make this claim without actually assessing their corresponding climatic outcome, or identifying how their assessment compares to current scientific standards (see points below).
- For the first time, a transparent assessment framework to calculate the climatic outcomes of short-term energy and emission scenarios is made openly available, reflecting the advances in open-source modelling tools that have been published after IPCC SR1.5. This allows the broader research community, and institutional modellers to apply such an assessment framework to make a consistent claim on the climatic outcome of their scenarios, and to indicate method choices (for instance, related to emission infilling, or harmonization, or choice of simple climate model) that could have result in their climate assessment differing from other estimates.
- The SR1.5, while noting the importance of assessing models (and scenarios) beyond Integrated Assessment Models, takes the self-assessed climatic outcomes of institutional scenarios as a given. As we demonstrate in this work, this does not necessarily facilitate a “like-for-like” comparison (especially given we find only two scenarios that would be appropriate to compare to other 1.5°C scenarios – the BP Net Zero and IEA Net Zero scenarios). The application of such a framework could allow for large scenario assessments, including the IPCC assessments, to broaden the types of models and scenarios included.

Specific comments:

in paragraph of line 74: there are 3 categories mentioned and only two are explained.

We have defined and distinguished between the three different types of scenarios as categorized by Skea et al.

The authors mention the carbon the carbon budget concept (paragraph of lines 137) but do not define carbon budget. Although the reader assumes the same calculation as the SR1.5 was taken, this should be clearly stated. e.g. From which to which year and which gases and sectors are included.

We have now included a definition of a carbon budget at the first instance where the term is used in the text, and have included text to point readers to the Supplementary Table that contains information on the gases necessary for the climate assessment.

With respect to your comment on the time horizon (years) considered in this analysis, we believe we have clearly indicated a need for an end-of-century (2100) multi-gas pathway, and have already indicated the existing time horizons of the scenarios explicitly in Table 1 (column “Scenario endpoint”).

Please discuss better the reason behind the statement in lines 106.

We have added a sentence pointing readers to the self-assessed claims presented by the institutional scenarios in Table 1.

The authors seem to consider IPCC and MAGICC6 as the gold standard of climate temperature calculations. While this is, generally, accepted in this literature, in order to make strong statements about institutional scenarios the authors should dedicate some discussion on why this model is better than any other that may have been used.

We have included additional text to the revised paper to discuss why we use MAGICC6 (to ensure backward compatibility with SR1.5, and consistent climate categorization).

You raise valid questions, which, we think to a large part is reflected in literature published since SR1.5 on the use of multiple simple climate models, and we have indicated a short discussion on this to bound the interpretation of our results.

However, the broader critique on the strength of statements applies as much to SR1.5, as it does to our assessment. We believe addressing this by using multiple simple climate models would be a novel contribution to the literature, but would be beyond the scope of this paper.

Still related to this point, in “Any reduced complexity climate model (e.g., FaIR21) which implements the openscm interface can be similarly applied.” Why this sensitivity was not performed? This would help the robustness of the statements made.

We have invited an additional author to our team for the revised manuscript, a developer of FaIR, and have performed this sensitivity check, and included a section in the Supplementary Information. We agree with the reviewer that this enhances the robustness of our conclusions and thank them for the insightful suggestion.

In Table S2, the row of “Energy and Industrial Process CO₂ (Mt CO₂ / yr)” is redundant as no infill method was used.

Two redundant rows have been removed, and the label has now been changed to “original” to reflect that this has not been infilled.

In Figure S1: should be given also in co₂ equivalent in order to better compare the impact in terms of warming contributions. The caption should mention what are the gray lines.

The purpose of this figure is to highlight the various emissions pathways arrived at by utilizing different infilling methods per gas. As it is directly generated from data from the SR15 dataset, we prefer to keep the figures in their original units per that data source. However, we agree that the origin was not clear and have added specific reference to it.

The authors decide on “no overshoot” term, which is miss leading and not coherent with the legends in the figures nor the results of figure 3.

The classification scheme used in this work is the same as that used in the IPCC Special Report on 1.5C, including the terminology in terms of “no overshoot” or “low overshoot” (termed “limited overshoot, less than 0.1C”). We agree that the application of the peak, and end of century warming probabilities were not clear, and have now updated the figure with grey dashed lines (see Panel C) to better map to the temperature categories introduced in previous sections.

I suggest a line on the zero in Figure 2. Also include the reduction relative to start year or the carbon budgets.

We have added a zero line as suggested, to better guide the eye of the reader. We believe that the presentation as it stands is the most clear visualization since, among other things, some of the emissions amounts become negative and wouldn't allow a clear evaluation of relative reductions.

A figure and discussion on total GHG would greatly help.

We have added a discussion on total GHG in the text as suggested. As you rightly point out, since we also refer to greenhouse gas neutrality as a condition for a scenario to be consistent with the Paris Agreement, this has helped link the initial sections of the text to the results better.

Although institutional scenarios probably not report on aerosol, a sensitivity on this is also required. Even if very simple, for example using the maximum reduction achieved with aerosol removal from the SR1.5 or SSP database (SSP may be better because it has different aerosol removal pathways).

Such a sensitivity is already included in the results from the original submission. Thank you for indicating that this was not clear. We have now expanded *Supplementary Table S2* to demonstrate an illustration of (selected) aerosol sensitivities.

Still related to figure 2, it would also add transparency to add which share of the emissions were calculated with the infill method.

We believe this indicator is not comparable across the different scenarios for two reasons:

- First, each scenario represents a different set of emission species;
- Second, different scenarios extend to different years (2040 or 2050)

It is the combination of these two factors that would be reflected in such a share of emission indicator, and since the comparison is not entirely "like-for-like", we would prefer not to include this share in the figures.

In "heavy reliance on land-based CDR that goes well beyond most of the low- and high overshoot pathways." Instead of "most" better to mention the quantile.

Thank you for the suggestion. We have now indicated the percentile in brackets after this statement in the revised paper.

In fig 4 total electricity is clearly not the best indicator to show.

We have now changed the indicator shown in the figure to capture the electrification of final energy consumption, which we believe is a more appropriate indicator to show.

I guess the number in brackets in the figures is the number of scenarios but this needs to be written in the caption. Please revise all the captions of your figures and make sure all the information is described in there.

We agree that this was unclear in the original submission, and has now been updated to be clearer.

Still related to Figure 4: without any indicator on negative emissions this figure may be very misleading. Same way the conclusion may also be less robust.

We don't understand why an indicator on negative emissions is necessary in this figure, which is considering different energy-system characteristics, and don't agree that this affects the robustness of the energy-system characteristics. However, there are two other areas in the paper, where we have addressed similar thematic issues, that we hope address your concerns include:

1. In the section of the paper "Comparing high-level mitigation levers", we discuss why we do not explicitly compare the negative emission indicators.
2. Based on your suggestion above, we have also included a discussion on total greenhouse gases.

Figure 4 seems to indicate that the institutional scenarios assume a very high energy demand that is

not foreseen by the models. Some models may reduce energy demand in the light of a temperature target, whereas institutionally scenarios foresee a much higher demand regardless of a climate constraint. This may be an important point of distinction between these scenarios. A discussion on total energy demand assumed would help the comparison (maybe taking only SSP5 like scenarios one would find the same ranges as the institutional ones).

We agree that the total (primary and final) energy demand can be an important indicator for how models differ, whether institutionally-based or from within the IAM families, and have now included final energy in Figure 4. Since our focus is on the temperature classification of the IAM scenario outputs to compare these with the institutional scenarios, we have not separated out different SSPs, another dimension that would not contribute to the aims of this paper.

Reviewer #2 (Remarks to the Author):

This study develops and applies a framework for examining whether alleged Paris-aligned climate scenarios of institutions such as Shell and IEA really are Paris-aligned. The study is timely, well-written and has the potential for a large impact, also beyond academia. I have a bunch of comments that mostly relate to clarity of presentation. I will be happy to recommend the study for publication after the authors have considered my (mostly minor) comments.

Thank you for your feedback on the relevance, and robustness of the study. We have taken on board your comments, which have improved the clarity of the manuscript.

My two only semi-major comments are that it is not easy to grasp the QRW infilling method (see detailed comments to line 129 and 508 below) and that the three criteria for being Paris-aligned that are presented in the introduction do not appear to be fully applied to evaluate the institutional scenarios later in the study (see detailed comment to line 184).

Please see the inline responses below.

Detailed comments:

- Line 18: If you have room in the abstract, I think you could broaden the appeal of your study by naming a couple of the institutions involved, such as Shell and International Energy Agency.

We have added this to the abstract.

- Line 39: As an example of the impact of institutional scenarios, you may want to refer to the recent ruling by The Hague District Court over Shell, where the court made several references to its Sky 1.5 scenario and its alleged alignment with the Paris goal:

<https://uitspraken.rechtspraak.nl/inziendocument?id=ECLI:NL:RBDHA:2021:5339>

We have added this to the introductory paragraph. Interestingly, the ruling also refers to two of the IEA scenarios we assess here (the SDS, and Net Zero scenario), in addition to the Sky scenario. Thank you for pointing us to this.

- Line 74: I encourage you to define what you mean by “outlook” and “exploratory” scenarios, as you do it for “normative” scenarios. I see that you partially do that around line 85, but I think the text might work better if you define all three terms upfront. Also, it is currently not clear from the text if you use “outlook” and “exploratory” interchangeably or if they mean different things.

We have defined and distinguished between the three different types of scenarios as categorized by Skea et al.

- Line 87-89: I am not sure I understand this sentence. What does large uncertainties relate to? Who are proposing the alternative scenarios (the creators of the outlook scenarios or you?). Consider rewording the sentence.

This has been reworded for clarity and to better connect to the discussion of scenario types above

- Line 99-101: What do you mean by “detailed greenhouse gas emission pathway”? Does it relate to a comprehensive coverage of GHGs and sources (as suggested by Table 1)? Please clarify.

The sentence has been reworded to clarify the meaning, which is that suggested by the reviewer.

- Line 117: It looks like the first step in Figure 1 is harmonization of historical and scenario data. Consider creating a better alignment between the figure and the text, for example by noting the step numbers on the framework figure.

We have implemented this suggestion by indicating the steps, and editing the text to improve clarity.

- Line 119-121: I was not familiar with the CQE method beforehand and had to consult your Methods section to understand the essence of it. I think this sentence could be improved if you add a bit of information, e.g.,: “with respect to a distribution of reference emissions scenarios that extends to the desired endpoint”. You could also consider mentioning that you used the scenario set from the Special Report on 1.5°C when applying the CQE method.

This recommendation has been included.

- Figure 1: Consider stating in the caption what the information in parentheses (Silicone, Aneris, etc.) refer to or delete it if it is not essential for the framework. Also, consider adding, to the captions, what the four different shapes/colours refer to.

We have removed the information in parentheses (these describe the open source modelling tools that we use to operationalise the step). Since these are described in the methods section, we’ve left them out of the figure. We have also added a legend to explain the different shapes in the figure.

- Line 129: Consider briefly (1-2 sentences) explaining the QRW infilling method, as you did for the CQE method.

This has now been done.

- Line 132-134: From reading this, I get the impression that you used the same modelling approach to assess the temperature outcomes of scenarios as in SR1.5? If so, consider stating that more explicitly. Also, please clarify what you mean by “classified”?

This has now been clarified in the text; it is a matter of comparing outcomes with the temperature conditions as given in Table S3 of the Supplementary Information and as used in SR1.5. For example, “1.5°C low overshoot” is defined by $0.50 < P1.5^{\circ}\text{C} \leq 0.67$ and $P1.5^{\circ}\text{C} (2100) \leq 0.50$.

- Table 1: The reference to the QRW infilling method seems out of place in the table caption. Why not mention “using MAGICC6”, as in Figure 3, instead?

Since different infilling methods were used, it is appropriate to mention QRW here. However, as the reviewer suggests, we have also mentioned MAGICC and given more detail about the contents of the table

- Figure 3: To create better alignment with the text and Table 1, you can consider showing the four pathways that you categorized to Lower 2°C in Figure 3a and the other two (better performing) pathways in Figure 3b.

This is a nice suggestion to improve the clarity of the figure, and we have updated the figure.

- Line 184: In the beginning of your study (lines 41-55), you present three climatic outcomes that scenarios should meet to be considered “Paris Agreement compatible”. However, in this paragraph, it appears that you are only addressing one of those three (whether scenarios are “very likely” to not exceed a peak warming of 2 degrees). For completion and consistency, I encourage you to explicitly comment on how the institutional scenarios perform relative to the other two criteria (“as likely as not” to limit warming to 1.5°C by the end of the century and achieving a balance between sources and removals). You could also indicate two of the criteria on Figure 3c (below 50% by 2100) and 3d (below 10% at all times).

We have now improved the text, in line with your suggestions. We include a discussion on the scenarios that achieve a balance between sources and sinks, and have updated Figure 3 to (visually) indicate the other two criteria.

- Line 200: perhaps something like “high-level mitigation levers” would be better than “technology-agnostic mitigation levers” in the heading. Otherwise, the choice of the “technology-agnostic” term only really makes sense after the reader has made it to the next section...

Text has been adjusted, as suggested.

- Line 210-212: If I understand correctly, you are here arguing why you did not consider CDR deployment and relative reduction in non-CO2 emissions? If so, I think you could present the argument more explicitly.

Yes, this is indeed the argument we were making, and we have rephrased it to try to make it more explicit

- Figure 4: The sequence of the charts does not match the figure caption (or the body text).

Both have now been fixed

- Line 447-449: I am not sure I understand. Does this, for example, mean that the emissions in all years of a scenario are downscaled a bit if the scenario's 2010 emissions are a bit higher than actual 2010 emissions? Consider rephrasing. ☞ After reading further, I see that the harmonization approach is detailed in lines 461-466. It may create a better reading experience if you only describe the harmonization approach once.

Agreed; this has now been edited

- Line 449: Missing left-hand parenthesis?

Extra right-hand parenthesis eliminated

- Line 454: Please provide a reference to the mentioned IEA projection.

This has been done.

- Line 464-465: Should it be “the default method of aneris” (its not like the software is an actor that can select things, right?)?

Yes, we have changed the text as suggested.

- Line 508: It would be helpful if you could give one or more short examples of how a certain characteristic of the trajectory of CO2 emissions from energy and industrial processes (the lead gas) results in a certain characteristic of the infilled gas trajectory, according to the QRW method. For example: Does a relatively high value for the lead gas also mean a relatively high value of the inferred gas? Does a steep reduction of the lead gas result in a proportionally steep reduction of the inferred gas? Or is the relationship more complicated?

These points have been clarified in the revised manuscript. Thank you for pointing out the need for further explanation.

REVIEWER COMMENTS

Reviewer #1 (Remarks to the Author):

The authors have made an effort to reply to my comments. I believe there is one that particularly still worries me:

"Still related to Figure 4: without any indicator on negative emissions this figure may be very misleading. Same way the conclusion may also be less robust.

We don't understand why an indicator on negative emissions is necessary in this figure, which is considering different energy-system characteristics, and don't agree that this affects the robustness of the energy-system characteristics. However, there are two other areas in the paper, where we have addressed similar thematic issues, that we hope address your concerns include:

1. In the section of the paper "Comparing high-level mitigation levers", we discuss why we do not explicitly compare the negative emission indicators.
2. Based on your suggestion above, we have also included a discussion on total greenhouse gases. "

In the discussion the authors mention the role of CDR in the energy system, and how institutional pathways lead to overshoot and high reliance on fossil (from lines 359). So I do not understand why the authors do not think negative emissions are important to change the energy system. Literature Realmonde et al. (2019) shows the impact of DAC availability on the energy system and the fact that it allows for more fossil. It maybe a possibility that institutional pathways rely on the expectation of large deployment of this technology, while IPCC scenarios implement normative bounds on the DAC growth capacity.

Reviewer #2 (Remarks to the Author):

The authors have done a convincing job at addressing my initial review comments. I only have a handful of additional comments, again mostly related to readability (all line numbers refer to the manuscript version with tracked changes):

- Abstract: Check journal requirements if you need to explain acronyms (BP and IEA).
- Introduction: I see that you have moved the "In this work we present..." paragraph down to just above the presentation of results. I am not convinced this works well. For example, the statement in line 97 "we assess scenarios from institutions including Shell" is now unclear (assess in what way?) and you now say that you will present (three) challenges to the assessment after you present them. I therefore advice you to consider a different structure (not necessarily as before, but the current structure does not work, in my opinion).
- Line 105-108: I think there could be a better link with the preceding text. What is the relevance to the issue of time horizon (I think I understand, but it could be clearer)? Why do you talk about outlook and exploratory scenarios when the previous paragraph identifies institutional scenarios as normative?
- Line 229: Since you illustrate your analysis of the other conditions for Paris consistency graphically, it might be nice to do the same for the "balance between sources and sinks in the second half of the century" condition. For example, you could plot the CO₂-eq emissions over time for the analyzed scenarios (the data currently "hidden" in Table S4). Also, I advice you to be explicit about the climate metric used to calculate the CO₂-eq emissions (GWP100, I assume?) and perhaps refer to recent discussions about the implication of the choice of climate metric for the goal of balancing sources and sinks, for example: <http://dx.doi.org/10.1098/rsta.2016.0445>.
- Figure 3d: For consistency with Figure 3c, consider inserting a horizontal grey line at 10%, indicating the threshold value for 'very likely less than 2°C' (>90% likelihood).

Please see our itemized responses to the comments by both reviewers. We would like to thank the reviewers for the thoroughness of their review of our original, and revised manuscript. The original comments are in black, our responses are in red

REVIEWERS' COMMENTS

Reviewer #1 (Remarks to the Author):

The authors have made an effort to reply to my comments.

Thank you for your feedback. We are glad we were able to reply to your comments.

I believe there is one that particularly still worries me:

"Still related to Figure 4: without any indicator on negative emissions this figure may be very misleading. Same way the conclusion may also be less robust.

We don't understand why an indicator on negative emissions is necessary in this figure, which is considering different energy-system characteristics, and don't agree that this affects the robustness of the energy-system characteristics. However, there are two other areas in the paper, where we have addressed similar thematic issues, that we hope address your concerns include:

1. In the section of the paper "Comparing high-level mitigation levers", we discuss why we do not explicitly compare the negative emission indicators.
2. Based on your suggestion above, we have also included a discussion on total greenhouse gases. "

In the discussion the authors mention the role of CDR in the energy system, and how institutional pathways lead to overshoot and high reliance on fossil (from lines 359). So I do not understand why the authors do not think negative emissions are important to change the energy system. Literature Realmonte et al. (2019) shows the impact of DAC availability on the energy system and the fact that it allows for more fossil. It maybe a possibility that institutional pathways rely on the expectation of large deployment of this technology, while IPCC scenarios implement normative bounds on the DAC growth capacity.

Thank you for giving us an opportunity to further explain our approach on the matter of CDR.

One of the key reasons it is not possible to make such a comparison is because emission reductions and removals are not explicitly reported by all the pathways (the IEA Net Zero scenario is an exception), and most scenarios do not report land use emissions (Shell Sky is an exception). However, your point on the importance of a discussion on negative emissions is very valid. We have included text in the revised manuscript indicating the importance of reporting the necessary data transparently, and hope this is sufficient to address your concerns.

While we could have opted to compute the net negative emissions from AFOLU, and non-AFOLU CO₂, we decided against this, since these were quantities that we computed (with the methods described in the paper), as opposed to data reported by the scenarios.

Reviewer #2 (Remarks to the Author):

The authors have done a convincing job at addressing my initial review comments. I only have a handful of additional comments, again mostly related to readability (all line numbers refer to the manuscript version with tracked changes):

Thank you for the feedback and we are glad that we could address your initial concerns and comment.

- Abstract: Check journal requirements if you need to explain acronyms (BP and IEA).

This has been addressed in the revised manuscript in line with journal requirements.

- Introduction: I see that you have moved the “In this work we present...” paragraph down to just above the presentation of results. I am not convinced this works well. For example, the statement in line 97 “we assess scenarios from institutions including Shell” is now unclear (assess in what way?) and you now say that you will present (three) challenges to the assessment after you present them. I therefore advise you to consider a different structure (not necessarily as before, but the current structure does not work, in my opinion).

We have restructured the Introduction, leaving the “In this work...” paragraph as the final paragraph as requested by journal guidelines. The rest of the Introduction now flows better, presenting context and previous work, followed by the challenges identified and resolved by our work.

- Line 105-108: I think there could be a better link with the preceding text. What is the relevance to the issue of time horizon (I think I understand, but it could be clearer)? Why do you talk about outlook and exploratory scenarios when the previous paragraph identifies institutional scenarios as normative?

In rewriting the Introduction, we have also addressed this comment; thank you for pointing out the redundancy in our previous formulation.

- Line 229: Since you illustrate your analysis of the other conditions for Paris consistency graphically, it might be nice to do the same for the “balance between sources and sinks in the second half of the century” condition. For example, you could plot the CO₂-eq emissions over time for the analyzed scenarios (the data currently “hidden” in Table S4). Also, I advise you to be explicit about the climate metric used to calculate the CO₂-eq emissions (GWP100, I assume?) and perhaps refer to recent discussions about the implication of the choice of climate metric for the goal of balancing sources and sinks, for example: <http://dx.doi.org/10.1098/rsta.2016.0445>.

Thank you for the suggestion. We have included two panels to Figure 3, where we now plot the aggregated greenhouse gas pathways as suggested. We have also included a short discussion on the reasons for selecting GWP100 to aggregate the emissions.

- Figure 3d: For consistency with Figure 3c, consider inserting a horizontal grey line at 10%, indicating the threshold value for ‘very likely less than 2°C’ (>90% likelihood).

We have added this in the revised manuscript (now, Figure 3e).